**Subject Category:**
Biology (whole organism)

ecology

vaquita, *Phocoena sinus*, passive acoustic monitoring, Bayesian modelling, population trend, wildlife management

**Author for correspondence:**
Armando M. Jaramillo-Legorreta
e-mail: ajaramil@cicese.mx

†Present address: Departamento de Biología de la Conservación, Centro de Investigación Científica y Educación Superior de Ensenada, Carretera Ensenada-Tijuana 3918, Zona Playitas, Ensenada, Baja California 22860, Mexico.

# Decline towards extinction of Mexico's vaquita porpoise (*Phocoena sinus*)

Armando M. Jaramillo-Legorreta[1], Gustavo Cardenas-Hinojosa[1,†], Edwyna Nieto-Garcia[1], Lorenzo Rojas-Bracho[1], Len Thomas[2], Jay M. Ver Hoef[3], Jeffrey Moore[4], Barbara Taylor[4], Jay Barlow[4] and Nicholas Tregenza[5]

[1]Comisión Nacional para el Conocimiento y Uso de la Biodiversidad, CICESE Camper 10, Carretera Ensenada-Tijuana 3918, Zona Playitas, Ensenada, Baja California 22860, Mexico
[2]Centre for Research into Ecological and Environmental Modelling, The Observatory, Buchanan Gardens, University of St Andrews, St Andrews, Fife KY16 9LZ, UK
[3]Alaska Fisheries Science Center, NOAA Fisheries, Marine Mammal Laboratory, Seattle, WA 98115, USA
[4]Southwest Fisheries Science Center, NOAA Fisheries, Marine Mammal and Turtle Division, 8901 La Jolla Shores Drive, La Jolla, CA 92037, USA
[5]Chelonia Limited, The Barkhouse, Mousehole TR196PH, UK

AMJ-L, 0000-0002-2876-6599; LT, 0000-0002-7436-067X; JMVH, 0000-0003-4302-6895; BT, 0000-0001-7620-0736; JB, 0000-0001-7862-855X; NT, 0000-0002-3811-0059

The vaquita (*Phocoena sinus*) is a small porpoise endemic to Mexico. It is listed by IUCN as Critically Endangered because of unsustainable levels of bycatch in gillnets. The population has been monitored with passive acoustic detectors every summer from 2011 to 2018; here we report results for 2017 and 2018. We combine the acoustic trends with an independent estimate of population size from 2015, and visual observations of at least seven animals in 2017 and six in 2018. Despite adoption of an emergency gillnet ban in May 2015, the estimated rate of decline remains extremely high: 48% decline in 2017 (95% Bayesian credible interval (CRI) 78% decline to 9% increase) and 47% in 2018 (95% CRI 80% decline to 13% increase). Estimated total population decline since 2011 is 98.6%, with greater than 99% probability the decline is greater than $33\% \, \mathrm{yr}^{-1}$. We estimate fewer than 19 vaquitas remained as of summer 2018 (posterior mean 9, median 8, 95% CRI 6–19). From March 2016 to March 2019, 10 dead vaquitas killed in gillnets were found. The ongoing presence of illegal gillnets despite the emergency ban continues to drive the vaquita towards extinction. Immediate management action is required if the species is to be saved.

# 1. Introduction

The vaquita (*Phocoena sinus*) is a species of porpoise endemic to the northern Gulf of California, Mexico. Historically, its population has declined because of unsustainable bycatch in gillnets, and it is listed as critically endangered by the IUCN. Since about 2010, an illegal gillnet fishery for an endangered fish, the totoaba (*Totoaba macdonaldi*), has resurged throughout the vaquita's range.

While vaquitas are difficult and expensive to survey visually, they are readily detectable using acoustics because they produce a nearly continuous series of echolocation clicks. This makes them excellent candidates for passive acoustic monitoring to estimate trends in abundance. In 2011, a systematic set of 46 acoustic sampling locations were established within the Vaquita Refuge (figure 1), a no fishing zone, and these have been monitored for a period of two months between June and August each year since then. Analysis of data from 2011 to 2015 showed an estimated decline in acoustic activity of 34% yr$^{-1}$ (95% Bayesian credible interval (CRI) 48% decline to 21% decline; [1]). Based on preliminary results through 2014, the government of Mexico enacted in 2015 an emergency 2-year ban on gillnets throughout the species' range to prevent extinction, at a cost of US$74 million to compensate fishers [2].

The acoustic monitoring programme was designed to produce estimates of temporal trend, not absolute population size. To obtain a population size estimate, a combined visual and acoustic survey was conducted in October and November 2015 covering the entire area of the gillnet exclusion zone (figure 1; note that the acoustic component of this survey was independent of the summer acoustic monitoring programme). This produced an estimate of about 60 vaquitas (posterior median 59, 95% CRI 22–145) [2]. Acoustic monitoring of vaquita during summer has continued through 2018. The most recent published analysis used acoustic data up to 2016 and estimated a decline of 49% (95% CRI 82% decline to 8% increase) between 2015 and 2016 [3]. Combining this finding with the 2015 population survey results, Thomas *et al.* [3] concluded that approximately 30 (posterior mean 33, median 27, 95% CRI 8–96) vaquitas remained as of autumn 2016.

Here, we provide updated estimates based on the two most recent years of monitoring data and new visual observations that give the minimum number of living vaquitas in autumn 2017 and 2018 [4,5]. The quantitative analysis presented here uses the same 46 acoustic sampling sites monitored since 2011 and an analytical method used previously [1,3], except for a small extension to accommodate the new observational data.

# 2. Material and methods

## 2.1. Relevant aspects of vaquita biology

The vaquita is found only in turbid waters in the far northwestern Gulf of California, Mexico [6,7]. Their range has reduced as abundance has declined [2], being recently confined to a small area towards the west margin of Vaquita Refuge (figure 1, blue polygon). Life expectancy historically is thought to have been approximately 20 years, with sexual maturity at 3–6 years and single calves born in the spring every 1–2 years [5,8]. Given these demographic parameters, maximum annual population growth rate was thought to be 4% [9], but the recent evidence for potential annual calving could increase this to roughly 6% [5]. Vaquitas are typically found in groups of one to three individuals, with an average of 2; this has not changed in recent visual surveys [2]. Like other porpoises, vaquitas make only high frequency narrow band echolocation clicks in regular sequences known as click trains [10]. Click rate is relatively constant [1], facilitating the use of acoustic detection rates to estimate trends in abundance.

## 2.2. Acoustic data collection and processing

The acoustic monitoring design and analyses were described in Jaramillo-Legorreta *et al.* [1]. Here we provide a brief overview. A grid of vaquita click detectors (C-PODs, manufactured by Chelonia Ltd., Mousehole, Cornwall, UK; http://www.chelonia.co.uk) was deployed in summers of 2017 and 2018 during the same season and at the same 46 core monitoring sites as previous monitoring studies from 2011 to 2016 [1,3]. In 2017, all sites were marked with surface buoys to facilitate rapid retrieval and replacement of C-PODs. To avoid complete data loss at any station due to instrument failure or loss, C-PODs were retrieved and replaced approximately every three weeks. As in previous analyses, vaquita click trains were identified with the KERNO classifier (v. 2.044; software freely available at http://www.chelonia.co.uk/cpod_downloads.htm) and validated by experienced analysts. This

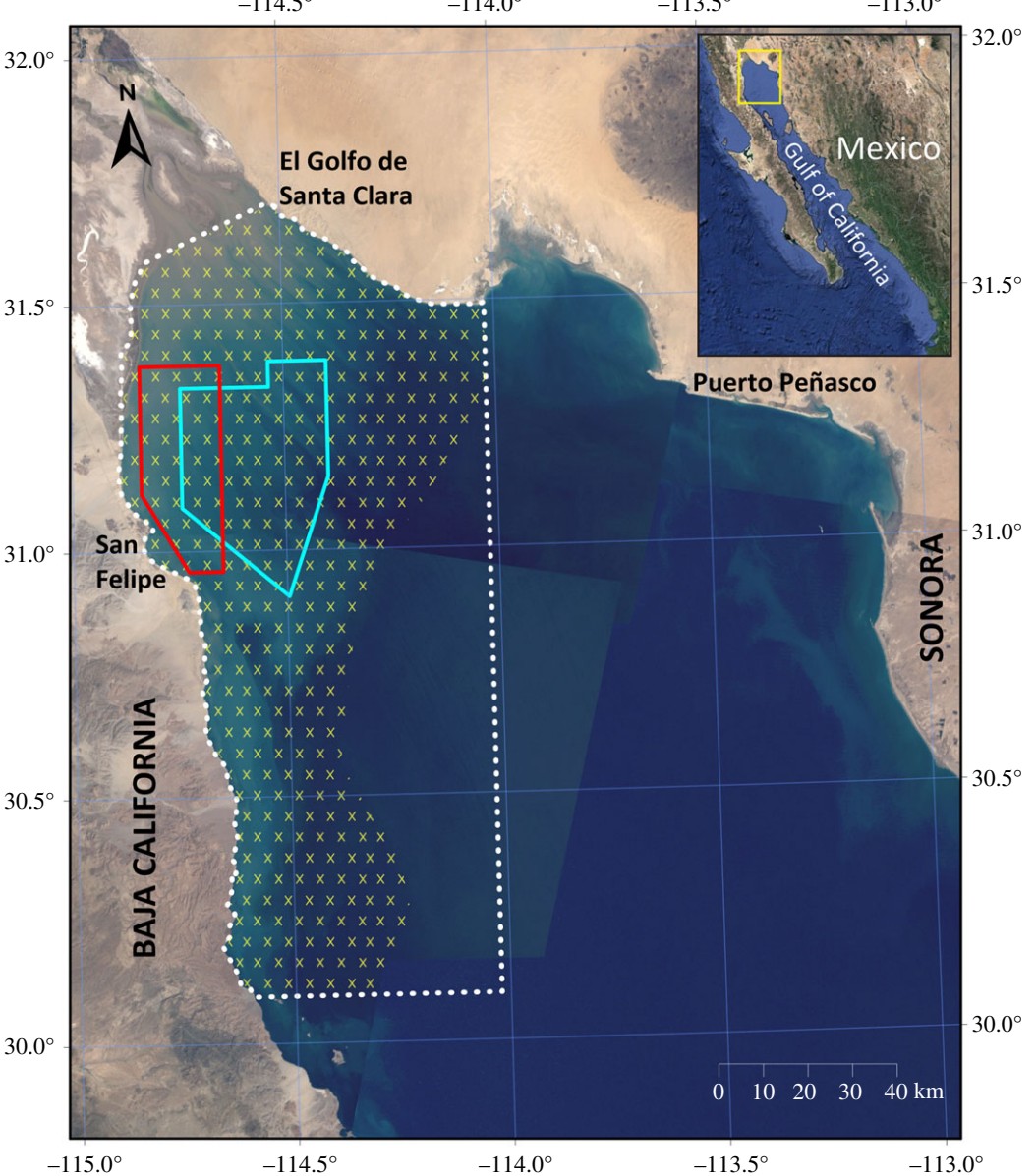

**Figure 1.** Historical distribution of vaquitas (yellow hatched area) in the northern Gulf of California. The Vaquita Refuge (agreed in 2005 and enforced in 2008 as a no fishing zone) is outlined in blue. The gillnet exclusion zone (where fishing with gillnets is banned but other types of fishing is allowed) was given straight boundaries (dotted white) described by single latitude and longitude to facilitate enforcement. An enhanced enforcement zone (red) was recommended by CIRVA in the area where the remaining vaquitas are thought to spend most of their time that also has high levels of totoaba fishing effort. Landsat satellite composite imagery provided by United States Geological Survey, National Aeronautics and Space Administration (NASA) and Esri, Inc. Projection UTM. Datum WGS84.

procedure results in a negligible level of false-positive detections, and detection rates that are not impacted significantly by variation in oceanographic conditions or acoustic behaviour of vaquitas [1,3]. Statistical analysis is based on data from the same 62-day period (19 June–19 August) in all years (dataset available in the electronic supplementary material). Trend estimates are based on the changes in the average number of vaquita clicks (in recognized click trains) per site per day. Detection positive minutes (DPMs, i.e. the number of minutes per day that contain one or more vaquita clicks) [11–13] are used as an index of vaquita abundance in figure 2.

## 2.3. Trend analyses

Previous analyses of the acoustic monitoring data [1,3] have used two statistical models—a geostatistical model and a non-spatial mixture model—to make inferences about trends in click rate over years. These

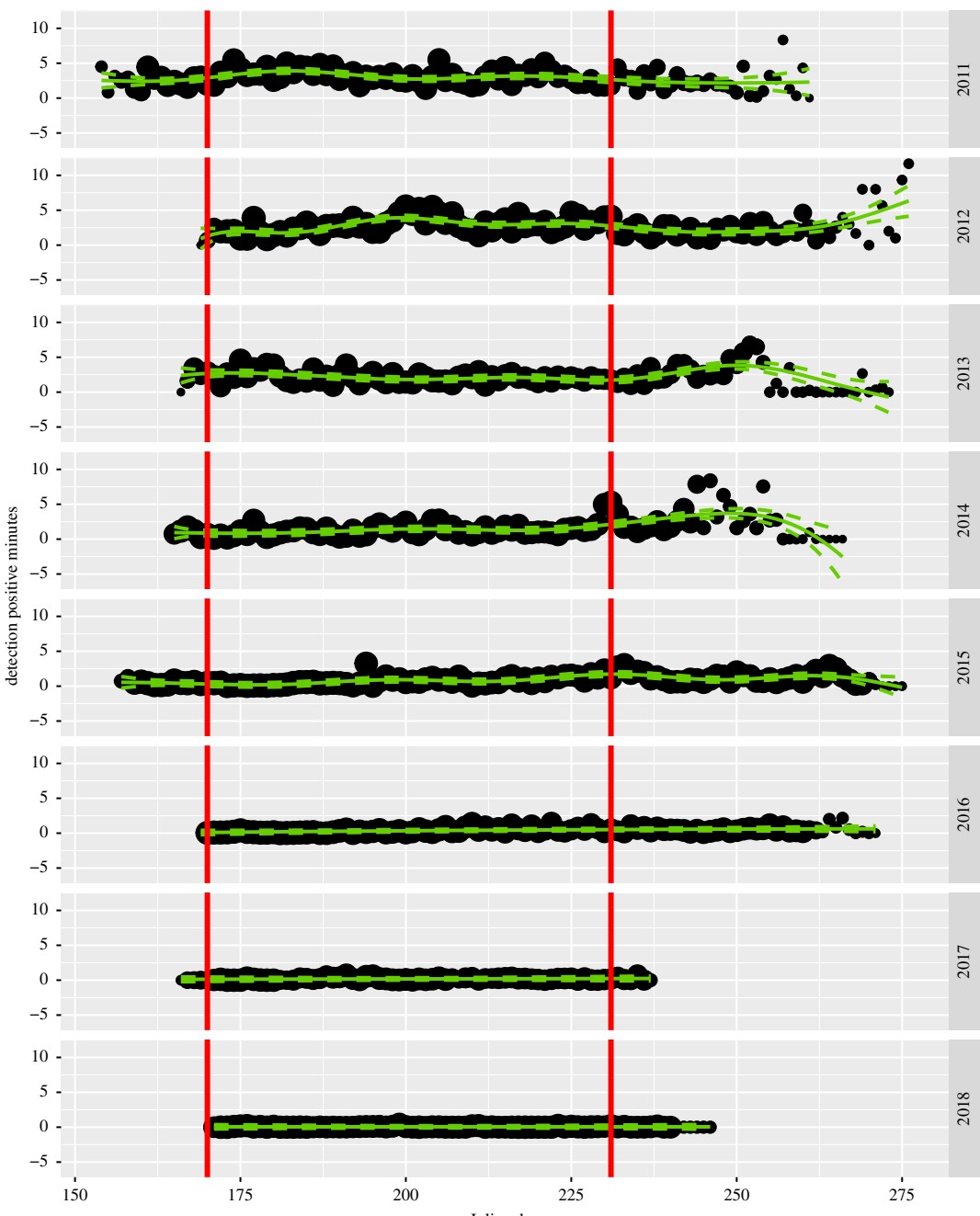

**Figure 2.** Mean acoustic detection positive minutes, averaged across the 46 core monitoring sites (*y*-axis) for each day of sampling (*x*-axis). Each dot represents a single day of sampling, with dot size proportional to the number of sites samples on that day. The green curves represent a smooth fit (a generalized additive mixed model with separate thin plate regression spline smooths per year, normal errors, identity link, weights that are number of sampling sites and auto-regressive error structure of order 1) with approximate 95% confidence interval shown as dashed lines. Vertical red lines indicate the core sampling period from Julian day 170–231.

models would not be necessary if sampling effort were balanced across C-PODs through time, but uneven sampling effort and missing data from some C-POD locations, mainly in the earlier years of the study, necessitate the model-based approach. Note that the models do not account for possible changes in the acoustic detection range of vaquita clicks or vaquita acoustic behaviour (see Discussion and conclusion). Here, we use only the geostatistical model because we found the mixture model is no longer a good fit to the data (see the electronic supplementary material). Despite this, we note that results from the mixture model are very similar to those from the geostatistical model, and results are essentially unchanged regardless of which is used (see the electronic supplementary material).

In brief, the geostatistical model compensates for locations with missing data by 'borrowing strength' from those around it: the model assumes the average click rate varies smoothly over space, with a separate smooth surface fit to each year of data but with the same level of smoothness (the spatial autocorrelation) across years. It further accounts for variation in sampling by assuming locations with more sampling days give more precise estimates of average click rate than those with fewer sampling days.

More details are given below; full model specifications are in Jaramillo-Legorreta *et al.* [1], while data and code are available in the electronic supplementary material. Let $W_{ti}$ denote the mean number of vaquita clicks detected in year $t$ at site $i$, averaged over $n_{ti}$ days of sampling. The data were log-transformed for analysis, $Y_{ti} = \log(W_{ti} + 1)$, and the resulting values are modelled as

$$Y_{ti} | \mu_t, Z_{ti}, \sigma_\varepsilon^2, n_{ti} \sim \text{Normal}\left(\mu_t + Z_{ti}, \frac{\sigma_\varepsilon^2}{n_{ti}}\right),$$

where $\mu_t$ is the expected mean clicks per day across sites in year $t$, $Z_{ti}$ is a spatially autocorrelated random effect and $\sigma_\varepsilon^2$ is the variance for spatially independent random error. The spatial random effect allows the number of clicks per day at each site within a year to depart from the overall mean, with sites in closer proximity to each other expected to have more similar departures from the overall mean. It took the form

$$\boldsymbol{Z}_t \sim \text{Multivariate normal}(0, \sigma_z^2 \boldsymbol{R}(\rho)),$$

where $\boldsymbol{Z}_t$ is the vector of site-specific random effects in year $t$, $(Z_{t1}, \ldots, Z_{t46})'$, $\sigma_z^2$ is the variance of the spatial random effect and $\boldsymbol{R}(\rho)$ is a $46 \times 46$ correlation matrix. The value for the $i$th row and $j$th column of $\boldsymbol{R}(\rho)$ is given by $\exp(-3h_{ij}/\rho)$, where $h_{ij}$ is the Euclidian distance (in kilometres) between sites $i$ and $j$, and $\rho$ is a parameter controlling the spatial smoothness of the random effect.

The model was fitted under the framework of Bayesian statistics. Uninformative prior distributions were used for all model parameters ($\mu_{2011}, \ldots, \mu_{2018}, \sigma_\varepsilon^2, \sigma_z^2, \rho$; see the electronic supplementary material). As a check, the model was re-run with wider prior distributions and near-identical results obtained. Samples from the posterior distribution were generated using Markov chain Monte Carlo (MCMC) methods via the OpenBugs software package [14]. One chain was used, with a mix of hand-chosen and randomly generated starting values (see the electronic supplementary material). Convergence was assessed using both Geweke's [15] and Heidelberger & Welch's [16] diagnostics, and (conservatively) the first 7500 samples were discarded as burn-in. Thereafter, we retained 1 000 000 samples (keeping every 100th sample to reduce the computational burden during post-processing)—this was sufficient to ensure at least three significant figure accuracy in posterior summaries. To check goodness-of-fit of the model, marginal predictive checks in the form of Bayesian *p*-values [17] were calculated for each site times year combination (see the electronic supplementary material).

The main outputs of interest from the model are annual changes in average acoustic activity, averaging over the sampling sites (but acknowledging the realized spatial variation, see [1]). These are given by

$$\lambda_{t,t+1} = \frac{B_{t+1}}{B_t},$$

where

$$B_t = \frac{1}{46} \sum_{i=1}^{46} (\exp(\mu_t + Z_{ti}) - 1).$$

Changes between any two time points can be calculated similarly—for example, the total change in average acoustic activity between 2011 and 2018 is given by $\lambda_{2011,2018} = B_{2011}/B_{2018}$. Values of $\lambda_{t1,t2}$ less than 1 indicate a decline; this is sometimes expressed as the percentage decline $(1 - \lambda_{t1,t2}) \times 100$.

## 2.4. Projected estimates of vaquita abundance

We assume that annual changes in acoustic activity reflect changes in vaquita population size (see Discussion and conclusion). This means that the estimated population size from the 2015 survey [2] can be projected forwards to give a population size in 2018 based on the estimated acoustic trends. The population abundance estimate from the 2015 survey ($\hat{N}_{2015}$) [2] can be represented by a lognormal distribution with mean 66 and standard deviation of 33. To project the population forward from 2015 to 2018, we drew 10 000 random samples from this lognormal distribution and multiplied

these in succession by 10 000 MCMC samples for $\hat{\lambda}_{2015,2016}$, $\hat{\lambda}_{2016,2017}$ and $\hat{\lambda}_{2017,2018}$. This generated population size estimates for each successive year ($\hat{N}_{2016}$, $\hat{N}_{2017}$ and $\hat{N}_{2018}$).

## 2.5. Updating estimates based on minimum count data

In October and November 2017, an effort was made to capture vaquitas [4]. A minimum of five vaquitas were observed through photographic identification, plus two different vaquitas were captured (i.e. at least seven known alive). In September 2018, an effort was made to obtain both photographs and biopsies from vaquitas. At one point, two different groups (one of four and one of two) were observed for a minimum of six individuals.

This information can be treated as new data for updating the posterior distribution of population sizes: with this additional information, the posterior probability of fewer than seven animals in autumn 2017 or six in autumn 2018 is zero. Hence, MCMC trajectories (where a trajectory consists of single set of MCMC draws for $\hat{N}_{2015}$, $\hat{\lambda}_{2015,2016}$, $\hat{\lambda}_{2016,2017}$ and $\hat{\lambda}_{2017,2018}$) for which the derived $\hat{N}_{2017}$ or $\hat{N}_{2018}$ were fewer than seven or six, respectively, were discarded. This resulted in retention of 2388 of the 10 000 samples mentioned previously. The retained truncated trajectories were used to generate updated posterior summaries of population size and trend between 2015 and 2018.

# 3. Results

## 3.1. Within-year patterns in acoustic data

The core monitoring period was 19 June–19 August each year (62 days). Acoustic data were sometimes missing during this period because of logger failure or loss; this could potentially bias annual trend estimates if there is a temporal trend within a monitoring period in instrument loss and also in acoustic detection rate. While there were some data losses for brief periods in 2017, generally sampling effort remained close to the maximum of 46 sensors throughout the monitoring period (figure 3). In addition, there was no strong pattern detected in the rate of vaquita detections within the monitoring period in any year (figure 2).

## 3.2. Annual change and trend analyses from acoustic data

The mean number of vaquita clicks detected per day, averaged over the sampling sites and days within the core monitoring period, decreased by 62.3% from 2016 to 2017 and by 70.1% from 2017 to 2018. In terms of annual change (i.e. the ratio of the value in year 2 divided by the value in year 1), the above declines in acoustic detections translate to $\hat{\lambda}_{2016,2017} = 0.377$ and $\hat{\lambda}_{2017,2018} = 0.299$. However, these values do not account for unequal effort across the sampling sites between years. The statistical models do account for unequal effort, in different ways, and also give estimates of uncertainty in the annual changes.

Results from the geospatial model are visually depicted in figure 4 and estimates of between-year change are given in table 1. Note the continuing range contraction first noted in Jaramillo-Legorreta *et al.* [1]. The values for years 2011–2016 are similar to those previously reported [1,3]. The posterior mean rates of decline in acoustic activity for the two new years are 49% decline from 2016 to 2017 (95% CRI 79% decline to 7% increase) and 58% decline from 2017 to 2018 (95% CRI 86% decline to 3% increase) (corresponding estimates of $\lambda$ are shown in table 1). The annual average decline between 2011 and 2018 has a posterior mean of 47% annually (95% CRI 54% decline to 40% decline), corresponding to a total decline of 99% over the 7-year period. While the actual rate of decline is uncertain, it is certain that the level of acoustic activity has declined since 2011 (posterior probability = 1), and there is a greater than 99% chance that the decline has averaged greater than 33% yr$^{-1}$. Moreover, the annual rate of decline seems to have increased over time, as evidenced by decreasing estimates of $\lambda$ (table 1).

The marginal predictive checks indicated no systematic departures of model predictions from data values (see the electronic supplementary material).

## 3.3. Population size and trend

Projecting forwards from the estimated population size in 2015 without accounting for the minimum count data, the posterior median estimate of population size in autumn 2018 (i.e. the end of the acoustic monitoring period) was just four animals. However, accounting for the seven animals seen in

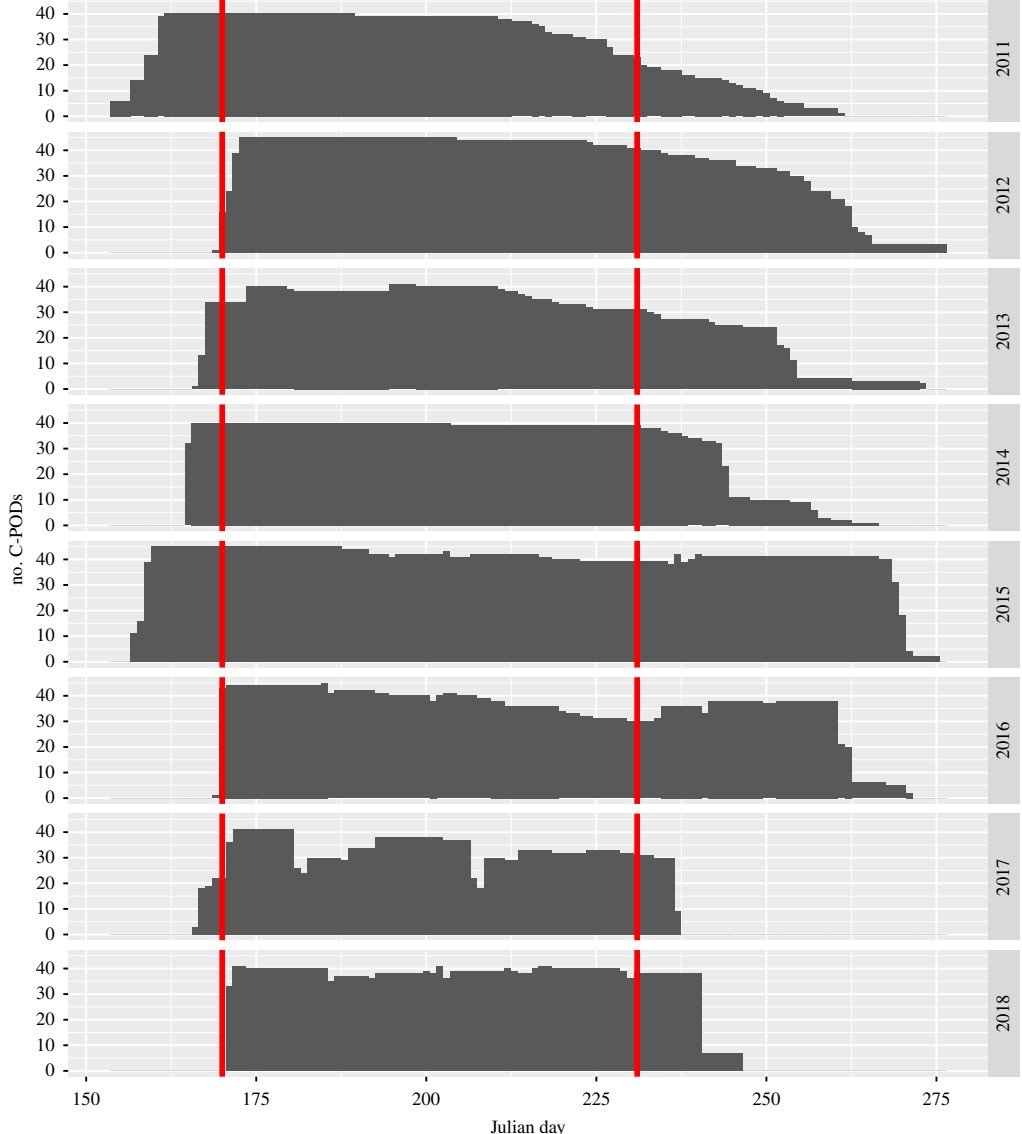

**Figure 3.** Number of acoustic loggers active in the 46 core monitoring sites by Julian day from 2011 to 2018. Vertical red lines indicate the core sampling period from Julian day 170–231.

2017 and six in 2018, the estimated population size was around nine (posterior mean 9, posterior median 8, 95% CRI 6–19). The full posterior distribution for each year is shown in figure 5 (together with historical estimates of population size, for context). We conclude that fewer than 19 vaquitas remained as of autumn 2018.

The knowledge of minimum known alive in 2017 and 2018 slightly changes the estimates of trend from the acoustic data (table 1), making them a little less negative. The posterior mean rate of decline is 48% in 2017 (95% CRI 78% decline to 9% increase) and 47% in 2018 (95% CRI 80% decline to 13% increase). However, the overall conclusion of a catastrophic long-term decline is unchanged: posterior mean total population decline since 2011 is 99%, with a probability of greater than 0.99 that this decline is greater than 33% $yr^{-1}$. There is no evidence that the decline slowed after the introduction of the gillnet ban in 2015: the posterior mean annual rate of decline from the periods 2015–2016, 2016–2017 and 2017–2018 is 45.8% (95% CRI 57.9% decline to 36.3% decline).

The minimum numbers known alive also retrospectively inform the 2015 estimate. For example, we can say that given there being at least seven animals alive in 2017 and six in 2018, then it is unlikely for the population size in 2015 to have been in the lower half of the estimated distribution reported by Taylor *et al.* [2]. Our updated estimate for the 2015 survey is a posterior mean of 100 vaquitas (posterior median 93, 95% CRI 47–191). This supersedes the value of ≈60 previously reported [2].

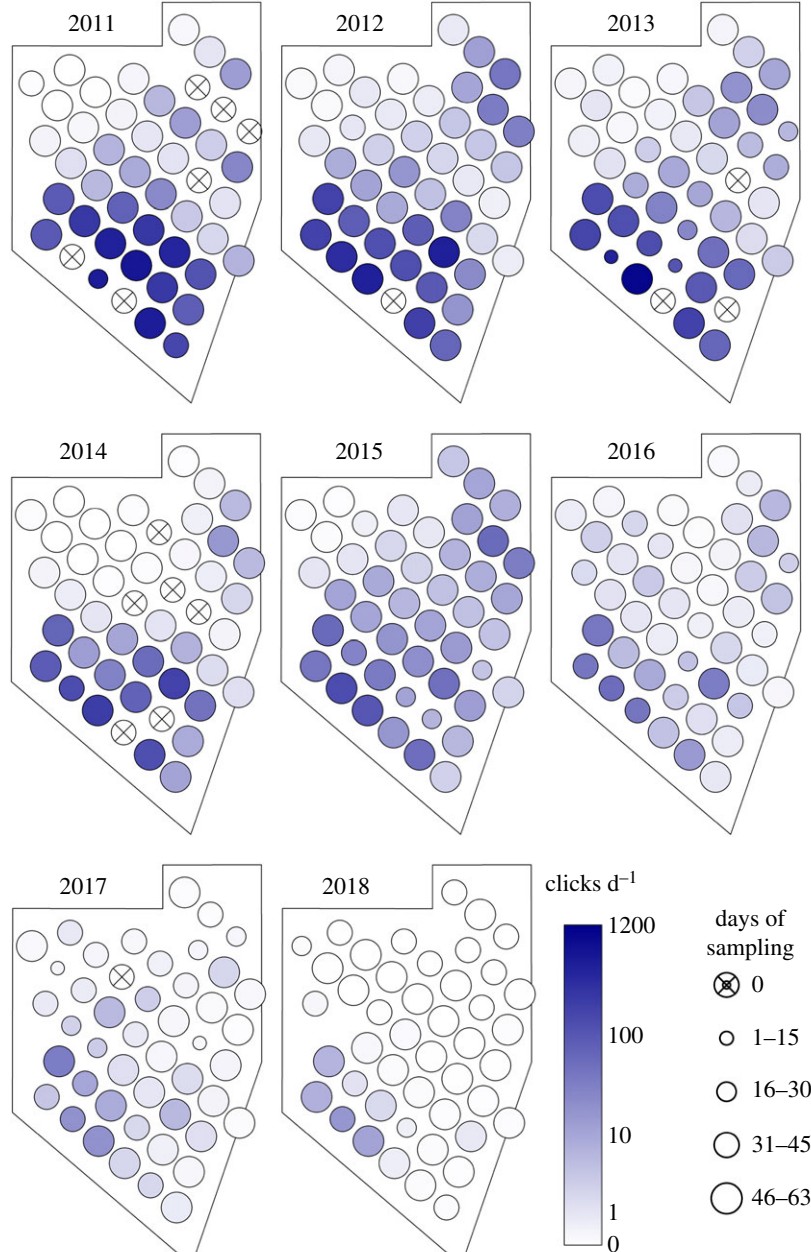

**Figure 4.** Estimated mean number of clicks per day predicted by the geostatistical model for the 46 numbered sampling sites with data for at least 1 year. Values in legend are posterior medians (note log scale). Some sites, circles with crosses, were missing in the indicated year. Size of circles indicates the number of sampling days on each year (see legend).

## 4. Discussion and conclusion

From the acoustic data alone, using the statistical models, the estimated decline in detection rate since monitoring started in 2011 is 99%. Although this estimate should be more reliable than a raw count, because it corrects for missing data, the change in raw acoustic detection rate is remarkably similar: an average of 4.37 clicks were detected per sensor per day of monitoring in 2011, and 0.052 in 2018, a decline of 99%. This gives us confidence that the acoustic trend estimates are robust.

To infer that vaquita population trends match the acoustic trends, we must assume that the acoustic behaviour of the vaquita and the underwater sound propagation conditions have not changed over time. This was investigated by Thomas *et al.* ([3], the electronic supplementary material), who found no evidence for any changes large enough to bias the trend estimates. We must also assume that the proportion of the population within the Vaquita Refuge has not changed over time, an assumption that is not directly testable with the data to hand. Taylor *et al.* [2] found that approximately 20% of

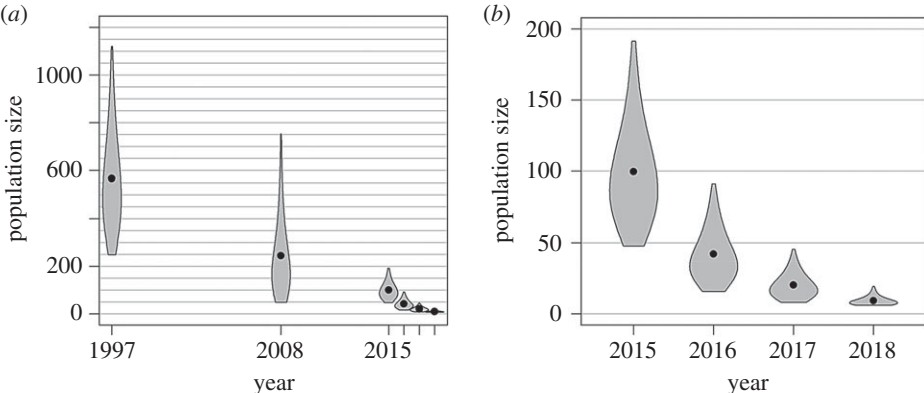

**Figure 5.** (*a*) Population size estimates from surveys conducted in 1997 [18], 2008 [19] and 2015 [2], and projected population size for 2016–2018. (*b*) Violin plots from 2015 onwards. Violin plots depict 95% confidence or credible limits and posterior means.

**Table 1.** Estimated per-year change ($\lambda$) in acoustic activity from the geospatial trend model applied to acoustic monitoring data both before and after incorporation of the additional sightings of vaquita in 2017 and 2018. Quantities are posterior means with 95% posterior credible intervals in brackets.

| | before incorporation of 2017 and 2018 sightings | after incorporation of 2017 and 2018 sightings | probability declining (%) | probability declining >20% yr$^{-1}$ (%) |
|---|---|---|---|---|
| 2011–2012 | 0.67 (0.22–1.43) | 0.66 (0.22–1.43) | 88.2 | 74.5 |
| 2012–2013 | 1.18 (0.43–2.77) | 1.17 (0.44–2.74) | 46.6 | 28.2 |
| 2013–2014 | 0.49 (0.15–1.54) | 0.50 (0.16–1.18) | 95.0 | 88.9 |
| 2014–2015 | 0.65 (0.27–1.29) | 0.59 (0.24–1.13) | 94.9 | 84.9 |
| 2015–2016 | 0.41 (0.18–0.76) | 0.43 (0.20–0.81) | 99.2 | 97.2 |
| 2016–2017 | 0.51 (0.21–1.07) | 0.52 (0.22–1.09) | 96.5 | 89.7 |
| 2017–2018 | 0.42 (0.14–0.97) | 0.53 (0.20–1.13) | 95.1 | 87.6 |
| geometric mean per-year change | 0.53 (0.45–0.60) | 0.55 (0.47–0.62) | ≈100 | ≈100 |

the population (12 of the estimated 59) were outside the refuge area during the time of the 2015 survey, so population trends would have to be very different outside the refuge for this to affect the overall population trend. If anything, given the lower levels of protection outside the refuge, population trends there are likely to be even more negative. Thomas *et al.* [3] found few detections on additional hydrophones placed at the periphery of the refuge; given the numbers in our current population estimate, it seems most likely that the vaquita population is now almost exclusively restricted to the refuge area.

Given our knowledge that at least seven vaquitas were alive in 2017 and at least six in 2018, and assuming the acoustic trend matches the trend in vaquita population size, then we conclude there were probably more vaquita in 2015 than the original estimate suggested. If this is correct, then the original underestimation for 2015 could have been due to sampling error (noting that the revised 2015 estimate falls comfortably within the original CRI) or due to some negative statistical bias. Taylor *et al.* [2] include a lengthy discussion of potential biases in their supplementary material and each of these biases could have contributed some small amount to any underestimation.

We see clear evidence that vaquita continue to decline precipitously despite the gillnet ban. Gillnet use continues [20] (Comité Internacional para la Recuperación de la Vaquita (CIRVA) 2017, 2018 unpublished data). In the 2018 totoaba season, there were 400 active totoaba nets recovered by a combined effort of the Mexican government, Sea Shepherd Conservation Society, Museo de la Ballena y Ciencias del Mar and WWF-Mexico. Most of these were from within an area recommended by the Comité Internacional para la Recuperación de la Vaquita (CIRVA) for increased enforcement because of the overlap between the vaquita distribution and past totoaba gillnet recovery (figure 1). Three dead vaquitas were found in 2016, five more in 2017, one in 2018 and one in 2019 (after the abundance estimate given in this paper and, therefore, not accounted for in the 2018 abundance

estimate). Of these 10, cause of death could be determined for eight, and all these deaths resulted from entanglement in gillnets [20] (CIRVA 2017, 2018 unpublished data).

With at most 19 vaquitas remaining in August 2018 and with their distribution contracted to a small area where a high amount of illegal totoaba fishing has occurred in the past and is continuing in the 2018–2019 totoaba spawning season (typically from December through May, peaking in March), the primary hope for this species is to guard the remaining individuals during the totoaba season. A dual approach combines the permanent presence of enforcement in the middle of the vaquita distribution and the active removal of illegal gillnets from the area and provides the most direct and immediate chance of survival for the remaining individuals. In addition, providing access, training and support to develop legal alternatives for fishers requires a longer time frame but is critical for increasing compliance with the gillnet ban in local communities. An effort to photograph and potentially biopsy vaquitas occurred in September 2018 and found the animals to be in robust health with two calves and evidence that vaquitas could calve annually [5]. This finding gives optimism for recovery if the killing could be halted immediately.

Ethics. The main authors are Mexico's Federal Government employees and met all the requirements to carry on the field sampling, including coordination with authorities of the Upper Gulf of California and Colorado River Biosphere Reserve and Federal Attorney of Environmental Protection.

Data accessibility. The dataset supporting this article has been uploaded as part of the electronic supplementary material (Microsoft Excel file). The code used to implement and run the models is also included in the electronic supplementary material, as text files and R packages.

Authors' contributions. A.M.J.-L. coordinated field sampling design, prepared acoustic files for vaquita signals identification, prepared the dataset for modelling and participated in modelling discussions; G.C.-H. coordinated field efforts and helped in data analysis; E.N.-G. analysed data for vaquita signal identification; L.R.-B. coordinated the research group; L.T. ran the geostatistical model and implemented the abundance projection; J.M.V.H. reviewed and refined the geostatistical model; J.M. implemented and ran the non-spatial mixture model; B.T. reviewed the abundance projection and drafted the first version of the work; J.B. reviewed the modelling and statistical analyses; N.T. advised on acoustic data acquisition and identification of vaquita acoustic signals. All authors collaborated to draft the manuscript and gave final approval for publication.

Competing interests. N.T. is owner of Chelonia Ltd, which manufactures the acoustic detector device used in these surveys (C-POD). The authors declare no other competing interests.

Funding. Field research, equipment and analyses were funded by Mexico's Ministry of Environment and Natural Resources, World Wildlife Fund Mexico and Museo de la Ballena y Ciencias del Mar (Mexico). The authors' analysis time was provided by their respective institutions.

Acknowledgements. Different institutions supported this work. We are especially grateful to the Mexican Secretaría del Medio Ambiente y Recursos Naturales for continuing the acoustic monitoring programme, especially to Rafael Pacchiano and Adriana Michel; the U.S. Marine Mammal Commission for their support since the very early stages of the acoustic monitoring, in particular to T. Ragen, R. Lent and P. Thomas; O. Vidal and E. Sanjurjo, from World Wildlife Fund (WWF) Mexico, for their continued support. We thank Le Equipe Cousteau, The Ocean Foundation, Fonds de Dotation pour la Biodiversité, MAAF Assurances (Save Your Logo), WWF-US and Opel Project Earth. We also thank Southwest Fisheries Science Center, NOAA Fisheries, in particular L. Ballance for marshalling NOAA support and A. Henry for logistical support throughout. We express our sincere thanks to the Fideicomiso Fondo para la Biodiversidad, the Instituto Nacional de Ecología y Cambio Climático (INECC), Comisión Nacional de Áreas Naturales Protegidas (CONANP), and the Directorate of the Reserva de la Biósfera del Alto Golfo de California y Delta del Río Colorado, M. Sau. Many thanks to our field staff J. Osuna, P. Valverde, R. Arozamena, Mauricio Najera and all the fishers who deployed and recovered the equipment. Jay ver Hoef developed the original geostatistical trend model. We thank the associate editor, Ruth King, and three anonymous reviewers for their helpful comments on an earlier draft.

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
