## [Reviewer comments · Royal Society Open Science]

Review History

RSOS-190598.R0 (Original submission)

Review form: Reviewer 1

Is the manuscript scientifically sound in its present form?

Yes

Are the interpretations and conclusions justified by the results?

Yes

Is the language acceptable?

Yes

Is it clear how to access all supporting data?

Yes

Do you have any ethical concerns with this paper?

No

Have you any concerns about statistical analyses in this paper?

No

Recommendation?

Accept with minor revision (please list in comments)

Comments to the Author(s)

The paper offers the latest estimates of population size for the critically endangered vaquita, As such, it is an important piece of work that should be published in a timely manner if the results are to have any influence on policies that could offer the species a chance of survival.

The employed methods are statistically sound, even if in some cases fairly strong assumptions have to be made to overcome limitations of the data. These assumptions are discussed and justified in the paper.

The conclusions drawn are appropriate and the discussion makes recommendations for future actions.

The writing style, grammar etc are in need of improvement but I hope that the editorial team could support the authors in that.

Review form: Reviewer 2

Is the manuscript scientifically sound in its present form?

Yes

Are the interpretations and conclusions justified by the results?

Yes

Is the language acceptable?

Yes

Is it clear how to access all supporting data?

Yes

Do you have any ethical concerns with this paper?

No

Have you any concerns about statistical analyses in this paper?

No

Recommendation?

Major revision is needed (please make suggestions in comments)

Comments to the Author(s)

Review of "Decline towards extinction of Mexico's vaquita porpoise (*Phocoena sinus*)" by Jaramillo-Legorreta and colleagues.

Thank you for giving me the opportunity to review this important paper. The vaquita is on the

brink of extinction, and providing abundance and trend estimates is of paramount importance to alert decision makers and raise awareness in the public opinion.

I like the paper and found the analyses sound and relevant. Below are my comments for the authors, I hope they will find them useful.

Regards,

Page 1, line 35. One may wonder why the need to combine 2015 abundance estimates to get results for 2017 and 2018; the authors might want to add a few words to make that clear.

Page 1, lines 37-38. I really like the way the authors convey uncertainty.

Page 2, line 11. "approximately 30 vaquitas remained": is it mean or median?

Page 2, section 3. A section on the species is missing. Basic information about the vaquita's ecology (life expectancy, breeding, sociality, etc...) will help to evaluate whether the model is relevant to estimate population trends and abundance.

Page 2, lines 32-33. Are there any misidentification issues? Noise in the signal? If yes, how were these issues addressed? More details are needed I guess.

Page 2, lines 52-53. I wonder whether the assumption of same spatial autocorrelation across years is valid for a declining and small population - depending on the age/sex structure, and how it is affected by the decline, spatial organization might vary across years. This is probably where more details about the vaquita's biology would be helpful (see my previous comment).

Page 2, lines 57-60. I understand that the two models have been described elsewhere. Usually I would ask for repeating the description so that the reader does not have to read another paper to understand the methods, but the supplementary materials are of high quality and do the job, the authors should be congratulated on that. This being said, from the current verbal model description, I could not say whether imperfect detection was accounted for or not. Can the authors add a few words on that if relevant?

Page 3, line 12. 1,000,000 MCMC samples seems like a long run, was the convergence difficult to reach? Thinning every 100th sample suggests autocorrelation was high, was it not? How was convergence assessed?

Page 3, line 14. I do not understand the term "model-averaged" posterior distributions". Could the authors elaborate on that? What did you model-average, and how?

Page 3, subsection 3.2. How the quality of fit of the model to the data was assessed? One way to do it in a Bayesian framework is to use posterior predictive checks (see chapter 6 of Bayesian Data Analysis by Gelman and co-authors). I would like to see an evaluation of the quality of fit of the models to the data.

Page 3, lines 38-44. I'm skeptical about removing all the trajectories that did not fit what was observed. Shouldn't this information be incorporated as a prior instead (not sure how though)? How can one assess the quality of fit of the model if the outputs are altered a posteriori? If many trajectories are in contradiction with what was observed, doesn't it mean that the model should be revised? Although I appreciate the authors' transparency about this practice, I would not recommend it. I'd be happy to be proved wrong of course.

Page 4, subsection 4.3. I applaud the authors' efforts to focus on population trends rather than abundance estimates.

Page 5, lines 25-28. I wonder if and how these data could be used to derive an abundance estimate (for another paper of course!). In a capture-recapture framework, these would be what are called dead-recoveries, and combining them with live recaptures (of some sort, acoustic or not) usually help in getting a survival estimate closer to actual survival and, as a consequence, better abundance estimates.

Review form: Reviewer 3

Is the manuscript scientifically sound in its present form?

Yes

Are the interpretations and conclusions justified by the results?

Yes

Is the language acceptable?

Yes

Is it clear how to access all supporting data?

Yes

Do you have any ethical concerns with this paper?

No

Have you any concerns about statistical analyses in this paper?

No

Recommendation?

Accept with minor revision (please list in comments)

Comments to the Author(s)

This is an excellent paper, on a very important topic. I have only two substantive questions, and some suggestions on ways in which the presentation could be improved.

You may want to say a little more about how “uninformative” the priors were. Perhaps some simple testing of different priors to demonstrate that they really were uninformative.

Interesting that many trajectories were rejected, on the basis they resulted in implausibly low population sizes. Could the same be happening at the upper end, with implausibly high population sizes but it’s not clear that they are implausible?

Figure 1 is complex and difficult to interpret. It would be useful to simplify the figure and clarify the figure legend. For example, there is no need to include the US / Mexican border in both the main figure and the inset. No need for a scale in both miles and km. Remove the Landsat satellite imagery text from the figure itself and put this information in the figure legend. I would remove all the text boxes, except the place names, from the figure and just include in the figure legend text like “red lines indicate gillnet exclusion zone, blue outline = vaquita refuge”, etc. The figure legend for Fig. 1 needs to be extended to clarify that the vaquita refuge is a no fishing zone (as indicated in the main text of the paper) and that the gillnet exclusion zone only bans one fishing method. The main text of the paper needs to why the enhanced enforcement zone is so small and

only partially overlaps the vaquita refuge. It would be useful to very briefly touch on this in the figure legend as well. This will seem puzzling to anyone looking at the figure.

It looks like the reference “Thomas et al. (2017)” in Line 11 on page 2 was meant to be “Thomas et al. [3]”. The second Thomas et al. reference in line 58 on page 4, is followed by “[3, Supplemental materials]”. Presumably this is the same reference. Later in the paper, the term “Supplementary Material” is used. This is the more common term. In any case, it would be good to use one consistent term- either Supplemental or Supplementary.

Suggest removing the word “fall” from “fall 2015 survey”. All surveys were done at the same time of year, right? The word fall suggests that there were several surveys a year.

This statement is not quite clear: “We assume that annual changes in acoustic activity reflect changes in vaquita population size (see Discussion). This means that the estimated population size from the 2015 survey [2] can be projected forwards to give a population size in 2018 based on the estimated acoustic trends.” Perhaps just briefly re-iterate here that the 2015 survey was based on visual as well as acoustic data and after 2015 only acoustic data are available. It sounds like population sizes in years after 2015 are based on the trend in acoustic detections and the 2015 estimate. It might be best not to say “projected forwards” because this could be interpreted to mean that you are projecting into future years (e.g. 2020 onwards). Rather than “projecting” the paper uses acoustic and visual survey data from 2015, in combination with acoustic data since 2015, to estimate population sizes in 2016, 2017 and 2018. Just clarify the text to make it clear exactly what is being done. You may want to consider inserting a few sentences from the supplementary material into the main text, to help explain the method in a little more detail.

Presumably “mean number of vaquita detected clicks per day” should read “mean number of detected vaquita clicks per day”?

Decision letter (RSOS-190598.R0)

28-May-2019

Dear Dr Jaramillo-Legorreta

On behalf of the Editors, I am pleased to inform you that your Manuscript RSOS-190598 entitled "Decline towards extinction of Mexico's vaquita porpoise (*Phocoena sinus*)" has been accepted for publication in Royal Society Open Science subject to minor revision in accordance with the referee suggestions. Please find the referees' comments at the end of this email.

The reviewers and handling editors have recommended publication, but also suggest some minor revisions to your manuscript. Therefore, I invite you to respond to the comments and revise your manuscript.

- Ethics statement

- Data accessibility

<http://datadryad.org/submit?journalID=RSOS&manu=RSOS-190598>

- Competing interests

- Authors' contributions

- Acknowledgements

- Funding statement

Because the schedule for publication is very tight, it is a condition of publication that you submit the revised version of your manuscript before 06-Jun-2019. Please note that the revision deadline will expire at 00.00am on this date. If you do not think you will be able to meet this date please let me know immediately.

If your manuscript is newly submitted and subsequently accepted for publication, you will be asked to pay the article processing charge, unless you request a waiver and this is approved by Royal Society Publishing. You can find out more about the charges at

<http://rsos.royalsocietypublishing.org/page/charges>. Should you have any queries, please contact openscience@royalsociety.org.

on behalf of Professor Ruth King (Associate Editor) and Kevin Padian (Subject Editor)
openscience@royalsociety.org

Associate Editor Comments to Author (Professor Ruth King):

The manuscript has been reviewed by three referees and myself. All are in agreement that this is an important and very timely paper. The authors have provided a very clear description of the statistical treatment of the data - particularly in relation to prior specification and they have been very honest in relation to dealing with what is regarded as "implausible" trajectories. The reviewers have identified a few places where the manuscript could be improved mainly in relation to further clarifications/explanations, most notably with regard to dealing with the implausible trajectories and potential misidentification issues, or descriptions in relation to the species. Linked to this is also the issue of goodness-of-fit of the models fitted to the data. Addressing these issues should strengthen the paper further. Some care is also required in relation to the writing itself.

Reviewer comments to Author:
Reviewer: 1

Comments to the Author(s)

The paper offers the latest estimates of population size for the critically endangered vaquita, As such, it is an important piece of work that should be published in a timely manner if the results are to have any influence on policies that could offer the species a chance of survival.

The employed methods are statistically sound, even if in some cases fairly strong assumptions have to be made to overcome limitations of the data. These assumptions are discussed and justified in the paper.

The conclusions drawn are appropriate and the discussion makes recommendations for future actions.

The writing style, grammar etc are in need of improvement but I hope that the editorial team could support the authors in that.

Reviewer: 2

Comments to the Author(s)

Review of "Decline towards extinction of Mexico's vaquita porpoise (*Phocoena sinus*)" by Jaramillo-Legorreta and colleagues.

Thank you for giving me the opportunity to review this important paper. The vaquita is on the brink of extinction, and providing abundance and trend estimates is of paramount importance to alert decision makers and raise awareness in the public opinion.

I like the paper and found the analyses sound and relevant. Below are my comments for the authors, I hope they will find them useful.

Regards,

Page 1, line 35. One may wonder why the need to combine 2015 abundance estimates to get results for 2017 and 2018; the authors might want to add a few words to make that clear.

Page 1, lines 37-38. I really like the way the authors convey uncertainty.

Page 2, line 11. "approximately 30 vaquitas remained": is it mean or median?

Page 2, section 3. A section on the species is missing. Basic information about the vaquita's ecology (life expectancy, breeding, sociality, etc...) will help to evaluate whether the model is relevant to estimate population trends and abundance.

Page 2, lines 32-33. Are there any misidentification issues? Noise in the signal? If yes, how were these issues addressed? More details are needed I guess.

Page 2, lines 52-53. I wonder whether the assumption of same spatial autocorrelation across years is valid for a declining and small population - depending on the age/sex structure, and how it is affected by the decline, spatial organization might vary across years. This is probably where more details about the vaquita's biology would be helpful (see my previous comment).

Page 2, lines 57-60. I understand that the two models have been described elsewhere. Usually I would ask for repeating the description so that the reader does not have to read another paper to understand the methods, but the supplementary materials are of high quality and do the job, the authors should be congratulated on that. This being said, from the current verbal model description, I could not say whether imperfect detection was accounted for or not. Can the authors add a few words on that if relevant?

Page 3, line 12. 1,000,000 MCMC samples seems like a long run, was the convergence difficult to reach? Thinning every 100th sample suggests autocorrelation was high, was it not? How was convergence assessed?

Page 3, line 14. I do not understand the term "model-averaged" posterior distributions". Could the authors elaborate on that? What did you model-average, and how?

Page 3, subsection 3.2. How the quality of fit of the model to the data was assessed? One way to do it in a Bayesian framework is to use posterior predictive checks (see chapter 6 of Bayesian Data Analysis by Gelman and co-authors). I would like to see an evaluation of the quality of fit of the models to the data.

Page 3, lines 38-44. I'm skeptical about removing all the trajectories that did not fit what was

observed. Shouldn't this information be incorporated as a prior instead (not sure how though)? How can one assess the quality of fit of the model if the outputs are altered a posteriori? If many trajectories are in contradiction with what was observed, doesn't it mean that the model should be revised? Although I appreciate the authors' transparency about this practice, I would not recommend it. I'd be happy to be proved wrong of course.

Page 4, subsection 4.3. I applaud the authors' efforts to focus on population trends rather than abundance estimates.

Page 5, lines 25-28. I wonder if and how these data could be used to derive an abundance estimate (for another paper of course!). In a capture-recapture framework, these would be what are called dead-recoveries, and combining them with live recaptures (of some sort, acoustic or not) usually help in getting a survival estimate closer to actual survival and, as a consequence, better abundance estimates.

Reviewer: 3

Comments to the Author(s)

This is an excellent paper, on a very important topic. I have only two substantive questions, and some suggestions on ways in which the presentation could be improved.

You may want to say a little more about how "uninformative" the priors were. Perhaps some simple testing of different priors to demonstrate that they really were uninformative.

Interesting that many trajectories were rejected, on the basis they resulted in implausibly low population sizes. Could the same be happening at the upper end, with implausibly high population sizes but it's not clear that they are implausible?

Figure 1 is complex and difficult to interpret. It would be useful to simplify the figure and clarify the figure legend. For example, there is no need to include the US / Mexican border in both the main figure and the inset. No need for a scale in both miles and km. Remove the Landsat satellite imagery text from the figure itself and put this information in the figure legend. I would remove all the text boxes, except the place names, from the figure and just include in the figure legend text like "red lines indicate gillnet exclusion zone, blue outline = vaquita refuge", etc. The figure legend for Fig. 1 needs to be extended to clarify that the vaquita refuge is a no fishing zone (as indicated in the main text of the paper) and that the gillnet exclusion zone only bans one fishing method. The main text of the paper needs to why the enhanced enforcement zone is so small and only partially overlaps the vaquita refuge. It would be useful to very briefly touch on this in the figure legend as well. This will seem puzzling to anyone looking at the figure.

It looks like the reference "Thomas et al. (2017)" in Line 11 on page 2 was meant to be "Thomas et al. [3]". The second Thomas et al. reference in line 58 on page 4, is followed by "[3, Supplemental materials]". Presumably this is the same reference. Later in the paper, the term "Supplementary Material" is used. This is the more common term. In any case, it would be good to use one consistent term- either Supplemental or Supplementary.

Suggest removing the word "fall" from "fall 2015 survey". All surveys were done at the same time of year, right? The word fall suggests that there were several surveys a year.

This statement is not quite clear: "We assume that annual changes in acoustic activity reflect changes in vaquita population size (see Discussion). This means that the estimated population size from the 2015 survey [2] can be projected forwards to give a population size in 2018 based on

the estimated acoustic trends." Perhaps just briefly re-iterate here that the 2015 survey was based on visual as well as acoustic data and after 2015 only acoustic data are available. It sounds like population sizes in years after 2015 are based on the trend in acoustic detections and the 2015 estimate. It might be best not to say "projected forwards" because this could be interpreted to mean that you are projecting into future years (e.g. 2020 onwards). Rather than "projecting" the paper uses acoustic and visual survey data from 2015, in combination with acoustic data since 2015, to estimate population sizes in 2016, 2017 and 2018. Just clarify the text to make it clear exactly what is being done. You may want to consider inserting a few sentences from the supplementary material into the main text, to help explain the method in a little more detail.

Presumably "mean number of vaquita detected clicks per day" should read "mean number of detected vaquita clicks per day"?

Author's Response to Decision Letter for (RSOS-190598.R0)

See Appendix A.

Decision letter (RSOS-190598.R1)

04-Jul-2019

Dear Dr Jaramillo-Legorreta,

I am pleased to inform you that your manuscript entitled "Decline towards extinction of Mexico's vaquita porpoise (*Phocoena sinus*)" is now accepted for publication in Royal Society Open Science.

on behalf of Professor Ruth King (Associate Editor) and Kevin Padian (Subject Editor)
openscience@royalsociety.org

Appendix A

Dear Professors Dunn, Padian and King,

Thank-you for your positive evaluation of our paper; thanks also to the three reviewers for their very helpful comments and suggestions. We have made revisions as requested and attach our resubmission. Below we reproduce the AE and reviewers' comments and give our responses in *blue italic* ink.

One reviewer comment, to undertake posterior predictive checks of both our models, caused us to realize that one of the models used (the non-spatial mixture model) is in some respects a poor fit to the data. We therefore removed this model from the main text into the supplementary materials, and base inferences in the main text only on the remaining (geospatial) model. The results from both models are very similar, so the main findings are almost unchanged. In making these changes, we have once again involved the originator of the geospatial model, Jay Ver Hoef (who was on our earlier papers analyzing previous years of data), and so we have added him as an author.

We very much hope the revised manuscript will be considered suitable for publication.

Respectfully yours,

Armando Jaramillo Legorreta and coauthors.

Associate Editor Comments to Author (Professor Ruth King):

The manuscript has been reviewed by three referees and myself. All are in agreement that this is an important and very timely paper. The authors have provided a very clear description of the statistical treatment of the data - particularly in relation to prior specification and they have been very honest in relation to dealing with what is regarded as "implausible" trajectories. The reviewers have identified a few places where the manuscript could be improved mainly in relation to further clarifications/explanations, most notably with regard to dealing with the implausible trajectories and potential misidentification issues, or descriptions in relation to the species. Linked to this is also the issue of goodness-of-fit of the models fitted to the data. Addressing these issues should strengthen the paper further. Some care is also required in relation to the writing itself.

We have revised the manuscript to deal with all of these issues, as detailed below. Given the changes needed after some reviewer comments, we are adding a co-author and re-arrange the order of them. We have also polished the English.

Reviewer: 1

Comments to the Author(s)

The paper offers the latest estimates of population size for the critically endangered vaquita, As such, it is an important piece of work that should be published in a timely manner if the results are to have any influence on policies that could offer the species a chance of survival.

The employed methods are statistically sound, even if in some cases fairly strong assumptions have to be made to overcome limitations of the data. These assumptions are discussed and justified in the paper.

The conclusions drawn are appropriate and the discussion makes recommendations for future actions.

The writing style, grammar etc are in need of improvement but I hope that the editorial team could support the authors in that.

We have been through the paper with a view to improving the style and grammar, and hope the editorial team agree things are now shipshape in that respect.

Reviewer: 2

Comments to the Author(s)

Review of “Decline towards extinction of Mexico’s vaquita porpoise (*Phocoena sinus*)” by Jaramillo-Legorreta and colleagues.

Thank you for giving me the opportunity to review this important paper. The vaquita is on the brink of extinction, and providing abundance and trend estimates is of paramount importance to alert decision makers and raise awareness in the public opinion.

I like the paper and found the analyses sound and relevant. Below are my comments for the authors, I hope they will find them useful.

Regards,

Page 1, line 35. One may wonder why the need to combine 2015 abundance estimates to get results for 2017 and 2018; the authors might want to add a few words to make that clear.

We added a sentence to the previous paragraph “The acoustic monitoring program was designed to produce estimates of temporal trend, not absolute population size.” before introducing the population size survey, so we hope this is now clear.

Page 1, lines 37-38. I really like the way the authors convey uncertainty.

Thank-you!

Page 2, line 11. “approximately 30 vaquitas remained”: is it mean or median?

It’s actually halfway between the posterior mean and median. We have added both, to be explicit:

“...concluded that approximately 30 (posterior mean 33, median 27, 95% CRI 8 to 96) vaquitas remained...”

Page 2, section 3. A section on the species is missing. Basic information about the vaquita’s ecology (life expectancy, breeding, sociality, etc...) will help to evaluate whether the model is relevant to estimate population trends and abundance.

Thanks for the suggestion. We added a new section at the start of Materials and methods and changed the subsequent section numbering accordingly. The new section reads:

“3.1 Relevant aspects of Vaquita biology

The vaquita is found only in turbid waters in the far northwestern Gulf of California, Mexico [6-7]. Their range has reduced as abundance has declined [2] being recently confined to a small area towards the west margin of vaquita refuge (figure 1, blue polygon). Life expectancy historically is thought to have been approximately 20 years, with sexual maturity at 3-6 years and single calves born in the spring every 1 to 2 years [5,8]. Given these demographic parameters, maximum annual population growth rate was thought to be 4% [9], but the recent evidence for potential annual calving could increase this to roughly 6% [5]. Vaquitas are typically found in groups of 1-3 individuals, with an average of two; this has not changed in recent visual surveys [2]. Like other porpoises, vaquitas make only high frequency narrow band

echolocation clicks in regular sequences known as click trains [10]. Click rate is relatively constant [1], facilitating the use of acoustic detection rates to estimate trends in abundance.”

Page 2, lines 32-33. Are there any misidentification issues? Noise in the signal? If yes, how were these issues addressed? More details are needed I guess.

Good question. We examined these issues extensively in previous works and found no evidence of such issues. We have added a sentence to be explicit about this:

“validated by experienced analysts. This procedure results in a negligible level of false-positive detections, and detection rates that are not impacted significantly by variation in oceanographic conditions or acoustic behavior of vaquitas [1,3].”

We also mention the assumptions required to go from acoustic to population trend in the Discussion.

Page 2, lines 52-53. I wonder whether the assumption of same spatial autocorrelation across years is valid for a declining and small population – depending on the age/sex structure, and how it is affected by the decline, spatial organization might vary across years. This is probably where more details about the vaquita’s biology would be helpful (see my previous comment).

We found no evidence of lack-of-fit of the geospatial model in any year (using posterior predictive checks, see below). We did add a section on vaquita biology, as mentioned earlier, although there is not enough biological knowledge to say how spatial organization might change as the population declines. Average group size does not appear to have changed, which may be one indication that behavioral spatial structure remains similar.

Page 2, lines 57-60. I understand that the two models have been described elsewhere. Usually I would ask for repeating the description so that the reader does not have to read another paper to understand the methods, but the supplementary materials are of high quality and do the job, the authors should be congratulated on that. This being said, from the current verbal model description, I could not say whether imperfect detection was accounted for or not. Can the authors add a few words on that if relevant?

The methods are designed merely to account for uneven sampling effort, and so give an accurate picture of changes in acoustic detection rates over time. Hence, we do not deal with imperfect detection at that stage. However, going from inferences about acoustic detection rates to inferences about population change does require assumptions about vaquita acoustic behavior and sound propagation. We have examined these assumptions in detail in previous papers, and we point to this in the second paragraph of the Discussion. Nevertheless, to be explicit in the Methods section, we added the following sentence after the lines indicated by the reviewer:

“Note that the models do not account for possible changes in the acoustic detection range of vaquita clicks or vaquita acoustic behavior – see Discussion.”

We also took on board the reviewer’s comment that they would typically ask for a repeat description of the model, and have added a brief mathematical description of the geospatial model into the main body of the paper.

Page 3, line 12. 1,000,000 MCMC samples seems like a long run, was the convergence difficult to reach? Thinning every 100th sample suggests autocorrelation was high, was it not? How was convergence assessed?

Convergence was assessed using both the Geweke and the Heidelberger & Welch diagnostics (geweke.diag and heidel.diag in the coda package in R). Both models actually converged quite quickly – our burn-in sample numbers were rather conservative (i.e., larger than necessary). In assessing the number of samples for inference, we aimed to get Monte Carlo error down to the level that the results we quote to 3 significant figures are repeatable (assessed using, e.g., batch means via the bm function in the batchmeans package). One million samples were more than sufficient and we

thinned only to make the summary computations quicker – autocorrelation was actually fairly low. The whole set of computations took only a few hours, but in hindsight, we could have used fewer samples and obtained almost identical results.

The text was expanded to reflect this: “One chain was used, with a mix of hand-chosen and randomly-generated starting values (see Supplementary Material). Convergence was assessed using both Geweke’s [15] and Heidelberger and Welch’s [16] diagnostics, and (conservatively) the first 7,500 samples were discarded as burn-in. Thereafter we retained 1,000,000 samples (keeping every 100th sample to reduce the computational burden during post-processing) – this was sufficient to ensure at least 3 significant figure accuracy in posterior summaries.”

Page 3, line 14. I do not understand the term ““model-averaged” posterior distributions”. Could the authors elaborate on that? What did you model-average, and how?

Sorry, this was opaque. However, we now base inference in the main paper on just one model, so this part is now redundant and we’ve deleted it.

In case the reviewer is interested, here was the rationale: we followed both previous papers where we used these models ([1] and [3]) by simply combining the two sets of 10,000 samples from each to make joint inference using both models. We did this in preference to any formal model selection or model averaging procedure in part because they are based on different response metrics (the geospatial model uses log-transformed data while the mixture model uses untransformed data) making likelihood-based comparisons difficult, and in part because we wanted results to be robust to model choice and not dominated by one or the other. In practice, the two produced rather similar results, particularly in the latter years where sampling was more complete.

Page 3, subsection 3.2. How the quality of fit of the model to the data was assessed? One way to do it in a Bayesian framework is to use posterior predictive checks (see chapter 6 of Bayesian Data Analysis by Gelman and co-authors). I would like to see an evaluation of the quality of fit of the models to the data.

This is a very good point! We did some checking when we first formulated these models but did not repeat it for the current paper. We therefore implemented marginal predictive checks, as suggested in the Gelman book (page 152 in the 3rd edition). The results for the geospatial model were very gratifying, but those for the non-spatial mixture model were the opposite! This is what led us to relegate that model to Supplementary Material.

What appears to be happening is that recent data violates the assumption of the mixture model that sites stay over all years in a fixed acoustic activity stratum (high, medium or low, relative to average activity in each year) – this is because the vaquita range has now contracted so much that areas were acoustic activity used to be relatively high now have no (or almost no) detections and so are in the “wrong” stratum. The mixture model does fit well at the level of aggregating sites within years (Bayesian p-values for data aggregated across all sites monitored each year are fine), and that the inferences from the mixture model are very similar to those from the geospatial model – but given the lack of fit we thought it better to remove it from the main paper. Thanks for helping us to spot this issue.

Page 3, lines 38-44. I’m skeptical about removing all the trajectories that did not fit what was observed. Shouldn’t this information be incorporated as a prior instead (not sure how though)? How can one assess the quality of fit of the model if the outputs are altered a posteriori? If many trajectories are in contradiction with what was observed, doesn’t it mean that the model should be revised? Although I appreciate the authors’ transparency about this practice, I would not recommend it. I’d be happy to be proved wrong of course.

We view the new observations, of a minimum abundance of 7 in 2017 and 6 in 2016, as additional data that can be used to update the posterior obtained from the acoustic monitoring data and the 2015 population survey. Having introduced these new data on minimum population size, MCMC samples that have a population size of less than these limits in the relevant years have a posterior probability of zero, and so can be removed from the set of valid samples. One could view

this as an importance sampling-based update with importance weights of zero attached to the impossible samples, and equal weights attached to all the other samples.

We have attempted to capture the idea better in the following text, although we don't explicitly mention that this is a form of importance sampling. We hope the reviewer finds this text better describes the motivation.

[Bottom of section 3.5] "This information can be treated as new data with which to update the posterior distribution of population sizes: with this additional information, the posterior probability of there being fewer than 7 animals in fall 2017 or 6 in fall 2018 is zero. Hence, MCMC trajectories (where a trajectory consists of single set of MCMC draws for N_{2015} , $\lambda_{2015-2016}$, $\lambda_{2016-2017}$ and $\lambda_{2017-2018}$) for which the derived N_{2017} or N_{2018} were fewer than 7 or 6, respectively, were discarded. This resulted in retention of 5,149 of the 20,000 samples mentioned previously. The retained truncated trajectories were used to generate updated posterior summaries of population size and trend between 2015 and 2018."

Page 4, subsection 4.3. I applaud the authors' efforts to focus on population trends rather than abundance estimates.

Thank-you.

Page 5, lines 25-28. I wonder if and how these data could be used to derive an abundance estimate (for another paper of course!). In a capture-recapture framework, these would be what are called dead-recoveries, and combining them with live recaptures (of some sort, acoustic or not) usually help in getting a survival estimate closer to actual survival and, as a consequence, better abundance estimates.

This is an excellent idea – thanks for the suggestion. Unfortunately, many of the carcasses found are too decomposed to allow matching with previous photos, and animals are not uniquely identifiable from their echolocation clicks. However, we plan to see whether we can do any matching on those animals that were not too badly decomposed.

Reviewer: 3

Comments to the Author(s)

This is an excellent paper, on a very important topic. I have only two substantive questions, and some suggestions on ways in which the presentation could be improved.

You may want to say a little more about how "uninformative" the priors were. Perhaps some simple testing of different priors to demonstrate that they really were uninformative.

Thank-you for the suggestion. As a check, we re-ran the models, doubling the ranges of the uniform priors for each parameter and obtained almost identical results. We added this to the text.

Interesting that many trajectories were rejected, on the basis they resulted in implausibly low population sizes. Could the same be happening at the upper end, with implausibly high population sizes but it's not clear that they are implausible?

Our current upper 95% Bayesian credible limit (as of fall 2018) is 22 animals. Given what we know of the situation in the Gulf this seems plausibly high, but unlikely – which is exactly what an upper credible limit should be.

Figure 1 is complex and difficult to interpret. It would be useful to simplify the figure and clarify the figure legend. For example, there is no need to include the US / Mexican border in both the main figure and the inset. No need for a scale in both miles and km. Remove the Landsat satellite imagery text from the figure itself and put this information in the figure legend. I would remove all the text boxes, except the place names, from the figure and just include in the figure legend

text like “red lines indicate gillnet exclusion zone, blue outline = vaquita refuge”, etc. The figure legend for Fig. 1 needs to be extended to clarify that the vaquita refuge is a no fishing zone (as indicated in the main text of the paper) and that the gillnet exclusion zone only bans one fishing method. The main text of the paper needs to why the enhanced enforcement zone is so small and only partially overlaps the vaquita refuge. It would be useful to very briefly touch on this in the figure legend as well. This will seem puzzling to anyone looking at the figure.

We have recomposed the map in figure 1 and the corresponding caption in agreement with your comments. However, we do disagree with your comment about the absence of explanation for the configuration of the “enhanced enforcement zone”. Actually, in page 5 lines 24-25, we say that the distribution of vaquita overlaps the distribution of totoaba illegal fishery. It is also said in the figure legend.

The caption now reads:

Figure 1. Historical distribution of vaquitas (yellow hatched area) in the northern Gulf of California. The Vaquita Refuge (agreed in 2005 and enforced in 2008 as a no fishing zone) is outlined in blue. The gillnet exclusion zone (where fishing with gillnets is banned but other types of fishing is allowed) was given straight boundaries (dotted white) described by single latitude and longitude to facilitate enforcement. An enhanced enforcement zone (red) was recommended by CIRVA in the area where the remaining vaquitas are thought to spend most of their time that also has high levels of totoaba fishing effort. Landsat satellite composite imagery provided by United States Geological Survey, National Aeronautics and Space Administration (NASA) and Esri, Inc. Projection UTM. Datum WGS84.

It looks like the reference “Thomas et al. (2017)” in Line 11 on page 2 was meant to be “Thomas et al. [3]”. The second Thomas et al. reference in line 58 on page 4, is followed by “[3, Supplemental materials]”. Presumably this is the same reference.

Thank you for finding this error. Corrected Thomas et al. (2017) to Thomas et al. [3].

Later in the paper, the term “Supplementary Material” is used. This is the more common term. In any case, it would be good to use one consistent term- either Supplemental or Supplementary.

We now use “Supplementary Material” throughout the document.

Suggest removing the word “fall” from “fall 2015 survey”. All surveys were done at the same time of year, right? The word fall suggests that there were several surveys a year.

There actually were two surveys in 2015: the summer acoustic monitoring and the fall population survey, which used visual and acoustic methods. To enhance clarity, we now say “2015 population survey”.

This statement is not quite clear: “We assume that annual changes in acoustic activity reflect changes in vaquita population size (see Discussion). This means that the estimated population size from the 2015 survey [2] can be projected forwards to give a population size in 2018 based on the estimated acoustic trends.” Perhaps just briefly reiterate here that the 2015 survey was based on visual as well as acoustic data and after 2015 only acoustic data are available. It sounds like population sizes in years after 2015 are based on the trend in acoustic detections and the 2015 estimate. It might be best not to say “projected forwards” because this could be interpreted to mean that you are projecting into future years (e.g. 2020 onwards). Rather than “projecting” the paper uses acoustic and visual survey data from 2015, in combination with acoustic data since 2015, to estimate population sizes in 2016, 2017 and 2018. Just clarify the text to make it clear exactly what is being done. You may want to consider inserting a few sentences from the supplementary material into the main text, to help explain the method in a little more detail.

The acoustic data used in the 2015 abundance estimate is completely independent of the yearly acoustic data used to estimate population trajectory since 2011. In 2015 we used acoustic data to cover shallow areas not navigable by the

oceanographic vessel. We use acoustic data collected in the areas navigated by the boat to calibrate against visual data, hence an estimate of abundance could be obtained in shallow areas. To make this clearer, when introducing the population survey at the beginning of the paper, we now say:

“The acoustic monitoring program was designed to produce estimates of temporal trend, not absolute population size. To obtain a population size estimate, a combined visual and acoustic survey was conducted in October and November 2015 covering the entire area of the gillnet exclusion zone (figure 1; note that the acoustic component of this survey was independent of the summer acoustic monitoring program)”

The yearly acoustic monitoring dataset is not useful to estimate abundance itself but indicates (with some assumptions) changes in abundance between years. Hence, we projected forward the 2015 abundance estimate using the estimated yearly changes obtained with the independent yearly acoustic estimates. We hope that this is clear now, given revisions made in the text to clarify the role of the population survey and the use of acoustic data in that survey.

Presumably “mean number of vaquita detected clicks per day” should read “mean number of detected vaquita clicks per day”?

Thank you for noticing this. Corrected.